# miRNA Predictors of Pancreatic Cancer Chemotherapeutic Response: A Systematic Review and Meta-Analysis

**DOI:** 10.3390/cancers11070900

**Published:** 2019-06-27

**Authors:** Madhav Madurantakam Royam, Rithika Ramesh, Ritika Shanker, Shanthi Sabarimurugan, Chellan Kumarasamy, Nachimuthu Ramesh, Kodiveri Muthukalianan Gothandam, Siddharta Baxi, Ajay Gupta, Sunil Krishnan, Rama Jayaraj

**Affiliations:** 1School of Biosciences and Technology, Vellore Institute of Technology (VIT), Vellore 632014, India; 2University of Adelaide, Adelaide, SA 5005, Australia; 3Genesis Cancer Care Centre, Bunbury, WA 6014, Australia; 4National Heart Institute, New Delhi 110065, India; 5Department of Radiation Oncology, Division of Radiation Oncology, The University of Texas MD Anderson Cancer Center, 1515 Holcombe Blvd, Houston, TX 77030, USA; 6College of Health and Human Sciences, Yellow 1.1.05, Ellengowan Drive, Charles Darwin University, Darwin, NT 0909, Australia

**Keywords:** pancreatic cancer, systematic review, meta-analysis, chemoresistance, miRNAs

## Abstract

Background: pancreatic cancer (PC) has increasing incidence and mortality in developing countries, and drug resistance is a significant hindrance to the efficacy of successful treatment. The objective of this systematic review and meta-analysis was to evaluate the association between miRNAs and response to chemotherapy in pancreatic cancer patients. Methods: the systematic review and meta-analysis was based on articles collected from a thorough search of PubMed and Science Direct databases for publications spanning from January 2008 to December 2018. The articles were screened via a set of inclusion and exclusion criteria based on the preferred reporting items for systematic review and meta-analysis (PRISMA) guidelines. Data was extracted, collated and tabulated in MS Excel for further synthesis. Hazard ratio (HR) was selected as the effect size metric to be pooled across studies for the meta-analysis, with the random effects model being applied. Subgroup analysis was also conducted, and the presence of publication bias in the selected studies was assessed. Publication bias of the included studies was quantified. Findings: of the 169 articles screened, 43 studies were included in our systematic review and 13 articles were included in the meta-analysis. Gemcitabine was observed to be the principal drug used in a majority of the studies. A total of 48 miRNAs have been studied, and 18 were observed to have possible contributions to chemoresistance, while 15 were observed to have possible contributions to chemosensitivity. 41 drug-related genetic pathways have been identified, through which the highlighted miRNA may be affecting chemosensitivity/resistance. The pooled HR value for overall survival was 1.603; (95% Confidence Interval (CI) 1.2–2.143; *p*-value: 0.01), with the subgroup analysis for miR-21 showing HR for resistance of 2.061; 95% CI 1.195–3.556; *p*-value: 0.09. Interpretation: our results highlight multiple miRNAs that have possible associations with modulation of chemotherapy response in pancreatic cancer patients. Further studies are needed to discover the molecular mechanisms underlying these associations before they can be suggested for use as biomarkers of response to chemotherapeutic interventions in pancreatic cancer.

## 1. Introduction

The GLOBOCAN 2018 data estimates that 18.1 million new cancer cases and 9.6 million cancer deaths occur worldwide. Pancreatic cancer (PC) accounts for 2.5% of the new cases and 4.5% of the overall death [1]. PC is the 12th most prevalent malignancy throughout the globe with 338,000 new cases recognised in 2012, and the five-year average prevalence rate was found to be 4.1 in 100,000 people throughout the world [2]. It has one of the lowest survival rates among the predominant tumours with a single digit five-year survival rate (2–9%) [3]. According to GLOBOCAN 2015 data, 1000 cases of PC are diagnosed daily [4], and 985 deaths [5] occur worldwide daily. With regards to India, cancer is one of the leading causing of death behind cardiovascular disease [6]. According to a study, the estimated annual PC burden in India in 2001 was 14,230 cases.

The mainstay of non-metastatic PC treatment is surgery [7]. Surgery is the most useful local treatment, but despite this, survival rates are modest primarily due to problems with distant and local recurrence issues. Therapeutic agents that are utilised as adjuvant or neo-adjuvant therapies are radiotherapy and systemic therapies (chemotherapies, targeted therapies). These include 5-Fluorouracil (5-FU) [8], 5-FU-based combinations like doxorubicin and mitomycin (FAM) [9], docetaxel [10], irinotecan [11], Gemcitabine (GEM) [12], irinotecan and oxaliplatin (FOLFIRINOX), GEM-based combination therapies such as 5-FU [13], capecitabine [14], S-1 [15], cisplatin [16], irinotecan [17], cisplatin, epirubicin and 5-FU (PEFG) [18], oxaliplatin [19], erlotinib [20] and nab-paclitaxel [21]. Other management options include nanocarriers [22], palliative care [23], immunotherapy [24].

MicroRNAs (miRNAs) are short (20–25 nucleotide) non-coding sequences increasingly recognised as molecular markers for the early detection of cancer and prognostication of clinical outcomes [25,26,27,28,29]. miRNAs regulate cell proliferating genes such as c-Myc and E2F1 thereby playing a critical role in cancer progression [30,31]. Differential expression of apoptosis-regulating genes was found to be associated with the 5-FU and GEM resistance in PC cell lines [32]. A tumour suppressing the role of miRNA was illustrated by the miR-96 miRNA which acts by suppressing KRAS gene in PC both in vitro and in vivo [33]. miRNA profiling by real-time PCR in PC cell lines and patient PC tissue samples revealed a range of 95 miRNAs that were altered [34]. Li Y et al. (2009) demonstrated that miRNA 200 and let-7a lead to the reversal of the epithelial-mesenchymal transition (EMT) in GEM-resistant pancreatic cancer cells [35]. A study by Zhu et al. observed that miRNA 27a and 451 were upregulated in multidrug-resistant cell lines (A2780DX5 and KB-V1) along with the overexpression of P-glycoprotein (P-gp) [36]. In the case of resistance due to the upregulation of miRNA-451, the transfection of miRNA-451 in doxorubicin-resistant MCF-7 cell lines increased the sensitivity to doxorubicin significantly [37]. 

Drug resistance is classified as intrinsic if it is present before treatment and acquired resistance if it develops while on treatment. There are several reviews involving miRNA and chemotherapy resistance [38,39,40]. Ali S et al. (2010) induced GEM sensitivity in PC cells through modulation of miR-200 and 21 by curcumin [41]. Reversal of EMT was achieved through natural agents (3,3′-diindolylmethane (DIM) or isoflavone) by upregulating miR-200 and let-7a in GEM-resistant PC cell lines [35]. The Notch signalling pathway is linked with the acquisition of EMT in GEM-resistant PC cell lines [42]. CD44-positive cancer stem cells (CSCs) were recently noted to be responsible for GEM-resistance in PC [43]. 

An increasing number of publications in recent years correlate miRNA expression in PC with resistance or sensitivity towards chemotherapeutic targets. Currently, the data on the correlations between PC chemoresistance/sensitivity and miRNA expression has not yielded clinically relevant solutions, in the form of prognostic biomarkers, despite the ongoing research in this field. A systematic review and meta-analysis approach allows us to collate the data across all published studies in the field and possibly highlight the associated miRNA, which may have clinical relevance in directing decisions regarding chemotherapy in PC patients [44,45,46,47]. Our study would help prospective scholars and clinicians by cataloguing miRNA alterations associated with chemotherapeutic response in this deadly disease. Future studies can then define their utility as predictors of chemotherapy response.

Our study aims to elucidate the relationship between miRNA expression and chemotherapeutic resistance or susceptibility in PC through a systematic review and meta-analysis of extant literature.

## 2. Methods

### 2.1. Search Strategy and Study Selection

Current studies in PC involving miRNA related drug resistance were identified through PubMed and Science Direct using the search terms; “microRNAs or miRNAs” AND “drug resistance” AND “PC” in combination. Four authors of this study (RJ, MRM, RR, and RS) independently performed a literature search systematically using the databases mentioned above (Appendix A). The study period was January 2008 to December 2018, inclusive. The search was limited to 10 years to make the information obtained relevant and contemporary. No language restriction was applied. The cross-references from the selected studies were searched for additional articles. Corresponding authors were contacted when the relevant information was not available in the publication. Discrepancies were resolved through discussion and consensus with a third reviewer.

### 2.2. Selection Criteria

The current study follows the preferred reporting items for systematic review and meta-analysis (PRISMA) guidelines [48]. The selection of studies was made based on the following inclusion and exclusion criteria.

### 2.3. Inclusion Criteria


Studies involving miRNAs expression and PC.Studies involving clinical patient data or preclinical data.Studies focusing on the resistance to some form of chemotherapy.Studies that reported the miRNA profiling platforms.Articles using in vitro assays to analyse the expression of miRNAs or gene related studies.


### 2.4. Exclusion Criteria


Studies published in non-English language and that do not involve drug resistance in PC were removed.Case reports, review articles, editorial, and studies with only in vitro or only PC patient samples data were excluded.


### 2.5. Data Extraction

MS Office Excel worksheet was used to collect information about studies that were included for extraction. Prior PRISMA guidelines were used to design the data sheet content. Full text and corresponding Appendix A of the following items were collected and recorded from the eligible studies; first author, year of publication, patient information, location of the study, ethnicity, gender, drug used, clinical stage, number of samples, lymph node metastasis, cell line(s) used, miRNA(s) involved, miRNA profiling platform, and drug pathways or genes associated.

### 2.6. Quality Assessment

The quality of eligible studies was assessed by two authors (RJ and MRM) critically according to the meta-analysis of observational studies in epidemiology (MOOSE) [49] for epidemiological studies by the predefined checklist. The qualifying studies had all the criteria either mentioned in the study or later denoted by the corresponding author (Appendix A).

### 2.7. Publication Bias

Two authors (R.J. and M.M.R.) assessed the risk of bias through a few distinct methods [47,50,51,52,53]. Egger’s and Begg’s bias indicator test was used to calculating publication bias along with an inverted funnel plot. Begg-Mazumdar bias indicator test was done to check the effect of publication and selection bias. Duval and Tweedie’s trim and fill calculation was calculated additionally to compute the effect size.

### 2.8. Meta-Analysis

A meta-analysis was performed for the obtained HR values and 95% confidence interval (CI) from the articles and Kaplan-Meier curves of the eligible studies using comprehensive meta-analysis (CMA) software version 3.0. Random effects models were used for meta-analysis. Cochran’s *Q* test and the *I*^2^ statistic [54] were performed to assess the statistical heterogeneity. If *p*-value > 0.05 heterogeneity was observed and the random effects model was performed. Forest plot was drawn to summarise the pooled HR estimate of the chemoresistance specific miRNAs. 

## 3. Results

Figure 1 explains our search selection and strategy in the form of a flowchart. The initial search resulted in 2671 studies from PubMed (*n* = 251) and Science Direct (*n* = 2420). After implementing the exclusion criteria, 169 articles were deemed relevant. After full-text screening and applying inclusion criteria, a total of 43 studies with miRNA expression related chemosensitivity or chemoresistance totalling 1963 individuals with PC was obtained for this study. The eligible articles were further reviewed (R.J., M.M.R.) and examined for data extraction (R.R. and R.S.). All the papers studied in our systematic review and meta-analysis were published in English. Out of the 43 studies, 23 were from China, seven were from the USA, seven were from Japan, five were from Germany, and one was from the Netherlands. Almost all studies (39 studies) used GEM as the primary drug for the treatment of PC. Both frozen and formalin fixed paraffin embedded (FFPE) tissue samples were used in the studies. Table 1 represents the descriptive characteristics of the included studies.

The most common assays performed is represented in Figure 2. A total of 59 cell lines have been used in the 43 studies, and PANC-1,2, Capan-1,2, HPDE and BxPc3 were the most commonly used ones. miRNA-21 modulates biological functions of PC cells including their proliferation, invasion, and chemoresistance studies used the highest number of cell lines (*n* = 14) [94]. 

In total, 48 miRNA have been studied in our systematic review; 23 of them were downregulated, and 25 were upregulated. In particular, nine upregulated miRNAs (15b, 17-5p, 21, 155, 181c, 203, 221, 320c and 1246) exhibited chemotherapeutic resistance and six upregulated miRNAs (21, 33a, 138-5p, 509-5p, 1207 and 1243) exhibited chemotherapeutic sensitivity. In contrast, nine downregulated miRNAs (7, 100, 124, 210, 200c, 205, 220b, 374b-5p and 497) exhibited chemotherapeutic resistance and nine downregulated miRNAs (101, 101-3p, 153, 203, 205-5p, 494, 506, 3656, let-7a) exhibited chemotherapeutic sensitivity. Four miRNA were differentially expressed. Overall, chemotherapeutic resistance (*n* = 18) and chemotherapeutic sensitivity (*n* = 15) were influenced by the miRNAs studied. The studies used GEM, lapatinib, capecitabine, 5-FU, a gamma-secretase inhibitor, Tarceva, radiation therapy, and AZD8055. 

Treatment with GEM led to the downregulation of miRNA 210 via the ABCC5 pathway, miRNA 124 via the polypyrimidine tract binding protein (PTBP1) and pyruvate kinase pathway, miRNA 103 via the ribonucleotide reductase M1 (RRM1) pathway, miRNA 100 via the FGFR3 pathway, miRNA 497 via the FGFR signalling pathway and miRNA 7 and 2015 via the class III b-tubulin (TUBB3) pathway; causing a chemoresistance phenotype. 

Treatment with GEM also led to the upregulation of miRNA 17-5p via the PTEN pathway, miRNA 221 via the EGFR1 and HER2 pathway, miRNA 203 via the activation of salt-inducible kinase (SLK1), miRNA 181c via the Hippo signalling pathway, miRNA 15b via the SMAD specific proteins pathway, miRNA 21 via the PTEN/Akt pathway, and VEGF, MMP-2 and MMP-9 proteins. Some studies noted upregulated miRNAs such as miRNA 221, 10a-5p and 21 no mechanistic pathways were identified. The upregulation of these miRNAs due to GEM treatment resulted in chemoresistance. 

GEM treatment also led to the downregulation of miRNAs, causing an increase in chemosensitivity, such as miRNA 3656 via EMT, miRNA let-7a via the HMGA2 pathway, miRNA 205-5p via the activation of K-Ras, Caveolin-1 and Ki-67, miRNA 153 via the SNAIL pathway, miRNA 101 via DNA-PKcs, miRNA 506 via the activation of NF-κB and SPHK1, miRNA 494 via SIRT1, c-myc pathway, miRNA 203 via the ZEB-1 pathway. GEM treatment upregulated some miRNAs causing an increase in chemosensitivities such as miRNA 509-5p and 1243 both via the E-cadherin pathway, miRNA 33a via the Akt/B-catenin pathway and miRNA 21 via the FasL/Fas pathway. 

5-FU is another drug that affects the regulation of miRNAs. Treatment with 5-FU led to the downregulation of miRNAs, thereby increasing chemoresistance, such as miRNA 200c and 200b both via directly targeting SUZ12, ROCK2. Treatment with 5-FU led to the upregulation of miRNA 1246 via the CCNG2 pathway, increasing chemoresistance. 5-FU also led to the downregulation of miRNA 494 via the SIRT1/c-myc pathway. Also, the upregulation of miRNA 138-5p via the vimentin resulted in chemosensitisation (Figure 3).

The miRNA and their drug targets based on the chemotherapeutic resistance and sensitivity is separated and listed out in Table 2 and Table 3 respectively.

## 4. Meta-Analysis

Associations between miRNA expression and patient survival were analysed using meta-analysis. The thorough screening revealed that 30 out of 43 studies did not report the HR values and 95% CI. Consequently, 30 studies were omitted from our meta-analysis due to insufficient reportage of data on patient survival and miRNA expressions. Collectively, 1088 patients from 12 studies were included in our meta-analysis (Figure 4). A pooled HR value of 1.603; 95% CI 1.2–2.143; *p*-value = 0.001 was obtained from a meta-analysis. Subgroup analysis between miR-21 showed an HR value of 2.061; 95% CI 1.195–3.556 and *p*-value of 0.009. Heterogeneity (*I*^2^ value) was observed to be 83.833 (Figure 5).

PC patients with elevated expression of miR-10a-5p (HR = 2.878, 95% CI = 1.614–5.131), 17-5p (HR = 1.18, 95% CI = 0.979–1.414), 21 (HR = 1.71, 95% CI = 1.147–2.535), 21 (HR = 0.32, 95% CI = 0.166–0.6), 33a (HR = 1.08, 95% CI = 1.002–1.168), 153 (HR = 1.16, 95% CI = 0.43–3.16), 181c (HR = 2.03, 95% CI = 1.33–3.11), 497 (HR = 2.76, 95% CI = 1.159–6.579), 506 (HR = 1.88, 95% CI = 1.048–3.026), 744 (HR = 21.2, 95% CI = 3.17–436) and 3656 (HR = 2.32, 95% CI = 1.37–3.57) exhibited chemotherapeutic resistance and a poorer prognosis. The low expression of miRNA-374b-5p (HR = 0.3, 95% CI = 0.17–0.54) revealed chemotherapeutic sensitivity and a better prognosis.

Figure 4 represents the forest plot of the primary meta-analysis of the pooled HR values along with the 95% CI from a pancreatic cancer patient, which are calculated using comprehensive meta-analysis (CMA) software (version 3.3.070, Biostat, Englewood, NJ, USA). The graphical representation of the right side of the plot is the HR, and 95% of the included studies and the red squares with the line represents the effect size of miRNA expressions. If the HR value is more than 1, it indicates the increased risk of the patient’s survival and less than specifies the decreased risk of patient’s survival. The size of the box indicates the weight of the study.

## 5. Publication Bias Indicators

### 5.1. Classic Fail-Safe N

This meta-analysis incorporates data from 12 studies, which yield a *z*-value of 6.26484 and the corresponding two-tailed *p*-value of 0.00000. The fail-safe N is 111. This means that we would need to locate and include 111 ‘null’ studies in order for the combined two-tailed *p*-value to exceed 0.050. Stated in another way, there would need to be 5.7 missing studies for every observed study for the effect to be nullified. 

### 5.2. Orwin Fail-Safe N

Here, the hazard ratio in observed studies is 1.150, which did not fall between the mean hazard ratio in the missing studies, so we could therefore not calculate the Orwin fail-safe N.

### 5.3. Begg and Mazumdar Rank Correlation Test

The Kendall′s tau b is 0.25758, with a one-tailed *p*-value of 1.16573 or a two-tailed *p*-value of 0.12186. This value compares the effect size and variance with the tau value and the value closes to 1 correlates to signify the publication bias.

### 5.4. Egger′s Test of the Intercept

In this case the intercept (B0) is 1.77800, 95% CI (−0.02031–3.57630), with *t* = 2.20298, df = 10. The 1-tailed *p*-value is 0.02609, and the 2-tailed *p*-value is 0.05218.

### 5.5. Duval and Tweedie′s Trim and Fill Test

Under the fixed effect model, the point estimate and 95% confidence interval for the combined studies are 1.15084 (1.07656, 1.23025). Using trim and fill, the imputed point estimate is 1.10505 (1.03516, 1.17967). Under the random effects model, the point estimate and 95% confidence interval for the combined studies are 1.60344 (1.19976, 2.14294). Using trim and fill, the imputed point estimate is 1.13206 (0.85429, 1.50016).

The hypothesis and heterogeneity testing are represented in Table 4.

Funnel plot is represented in Figure 6, and funnel plot with observed and imputed studies in Figure 7, which represents the possible bias between the included studies. If the studies have no publication bias, then the points will fall on the central line, indicating the symmetry. Every study or cohort included in the forest plot is represented as a point on the funnel plot. 

## 6. Discussion

Emerging studies have revealed that specific miRNA expression in PC is related to chemotherapeutic sensitivity/resistance and regulation of molecular signalling pathways. Therefore, a systematic review and meta-analysis to catalogue the miRNA expression patterns and decipher the relationships and regulation of oncogenic signalling pathways to outcomes would help in the prediction of chemotherapy response and triaging of future clinical therapeutic strategies in PC. 

We conducted this systematic review of published studies on miRNA expressions to inventory the full range of chemotherapeutic resistance and sensitivity in PC. To our knowledge, this is the first systematic review and meta-analysis describing the role of miRNA expression in chemotherapeutic resistance and sensitivity as well as molecular signalling pathways in PC. A total of 43 studies with 1963 patients were evaluated with 2207 PC tissue samples analysed. Also, 162 blood samples from the PC patients were examined in our study.

We found that distinct miRNAs were upregulated or downregulated in PC cell lines and tissues. In turn, they targeted specific molecular signalling pathways to mediate sensitivity or resistance to chemotherapeutic drugs in PC. Of the eight chemotherapeutic drugs studied individually and in combinations, GEM was the most studied followed by 5-FU. 

In a study on lung cancer cells, miR-17-5p downregulation was associated with an increased expression of beclin 1 gene, which is an autophagy modulator in the survival pathway [96]. Furthermore, this miRNA belongs to the miR-17-92 cluster and is upregulated in PC, where it is linked to pancreatic carcinogenesis [97,98]. Downregulation of miRNA 7 and 205 resulted in chemoresistance in PC, and a possible target is TUBB3, as noted in another study [99,100]. 

Our meta-analysis showed a non-significant pooled effect size, suggesting that drug-resistance specific miRNAs may not necessarily be good predictors of patient survival. However, it is essential to note that we have used only nine clinical studies in our meta-analysis due to a general lack of studies reporting clinical outcomes via HR and 95% CI values. From the data on publication bias, we concluded that several factors might be responsible for high heterogeneity between the data, such as study strategy, inadequate information and sample size [101]. The publication bias results computed that the studies required for analysing the statistically non-significant overall effect, and there were no missing studies as per the report [44,46,102].

The fact that many of the included studies did not investigate the HR values, exhibits a significant limitation in our quantitative synthesis [46]. Several included studies have investigated differentially expressed miRNA as drivers of drug resistance in PC and as prognostic markers by comparing their expression level in a tumour and adjacent healthy tissue.

## 7. Limitations and Strengths of Our Study

High variability in the data was observed and could be the result of factors such as (1) different sample groups; (2) diverse validation methods; and (3) limited sampling. Overcoming tumour resistance remains a significant challenge in the treatment of PC. The significant factors in chemo-resistance are the presence of miRNA expressions and their genetic regulation. Our systematic review and analysis attempted to show the association between the expression levels of miRNAs and chemotherapeutic resistance in PC. This review would help achieve a better understanding of the overall network of the mechanisms contributing to drug resistance and the regulation of the correlated miRNAs.

## 8. Conclusions

This systematic review and meta-analysis of miRNA expressions identified critical determinants of chemotherapeutic sensitivity and resistance in PC and correlated these with oncogenic molecular signalling pathways. The knowledge based on the miRNA and drug targets could drive the choice of chemotherapeutic regimens used in patients, adoption of combination therapies that overcome therapeutic resistance, utilisation of miRNAs as biomarkers of chemotherapy response, and early adaptive modifications to treatment in response to alterations in miRNA profiles of tumours. 

## Figures and Tables

**Figure 1 cancers-11-00900-f001:**
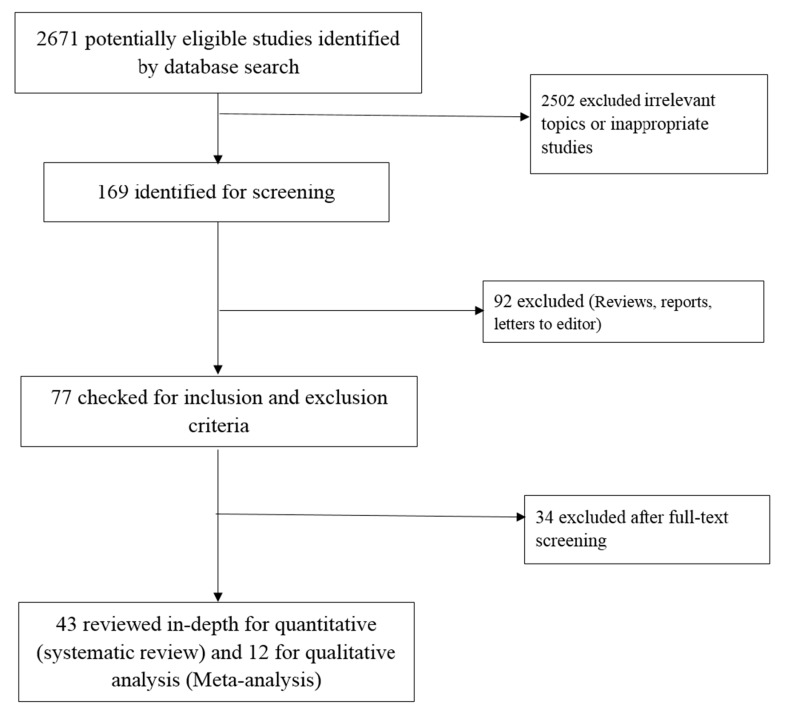
Flowchart of the literature study process and selection.

**Figure 2 cancers-11-00900-f002:**
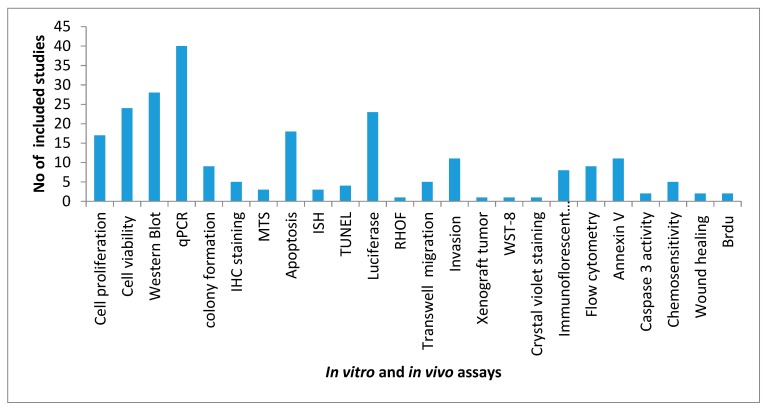
Commonly performed in vitro assays in the included articles. ISH: in-situ Hybridization; IHC: immuno histo-chemistry; TUNEL: terminal deoxynucleotidyl transferase (TdT) dUTP Nick-End Labeling.

**Figure 3 cancers-11-00900-f003:**
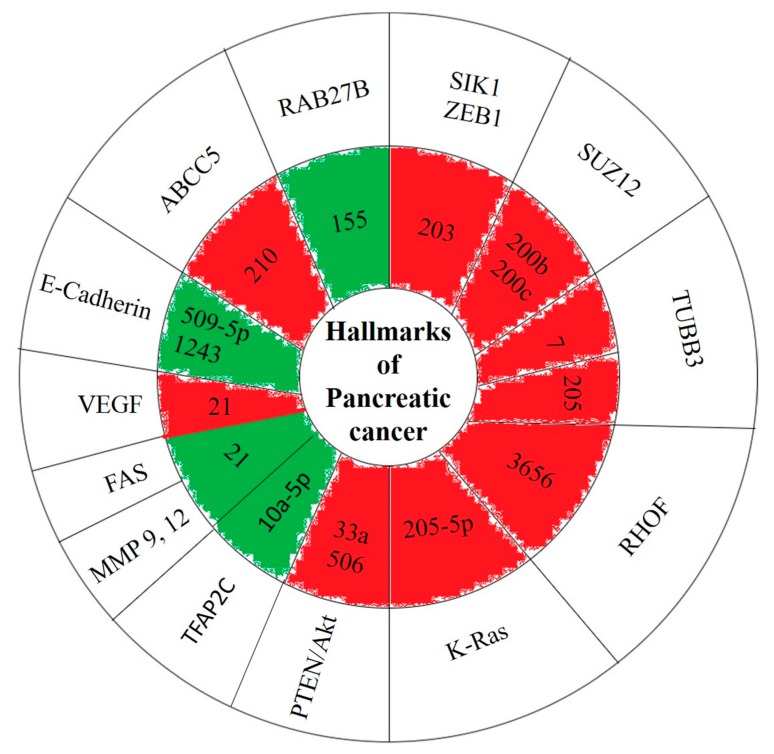
Hallmarks of PC. The shells highlighted in red are miRNAs that are downregulated while green denotes upregulated in pancreatic cancers in comparison to normal tissue.

**Figure 4 cancers-11-00900-f004:**
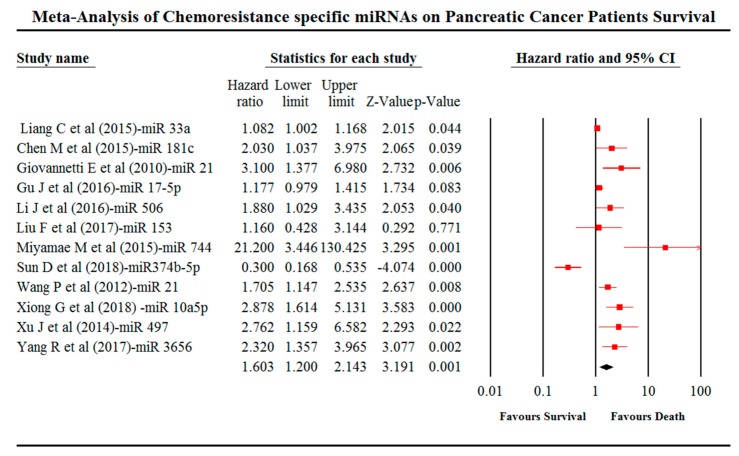
Forest plot for the HR and 95% CI of PC studies using meta-analysis. Random effect model; I squared = 83.833%; tau = 0.403; *Q* value = 68.042; df = 11.

**Figure 5 cancers-11-00900-f005:**
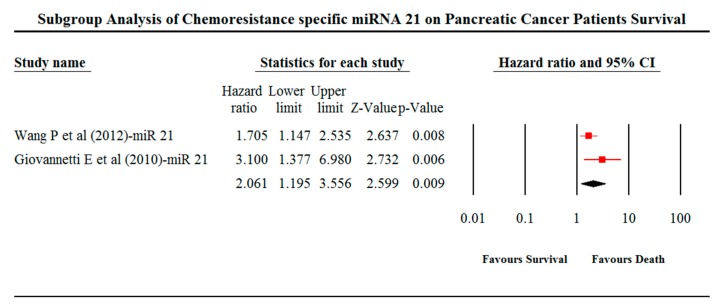
Forest plot of sub-group analysis for miRNA-21 of the included studies.

**Figure 6 cancers-11-00900-f006:**
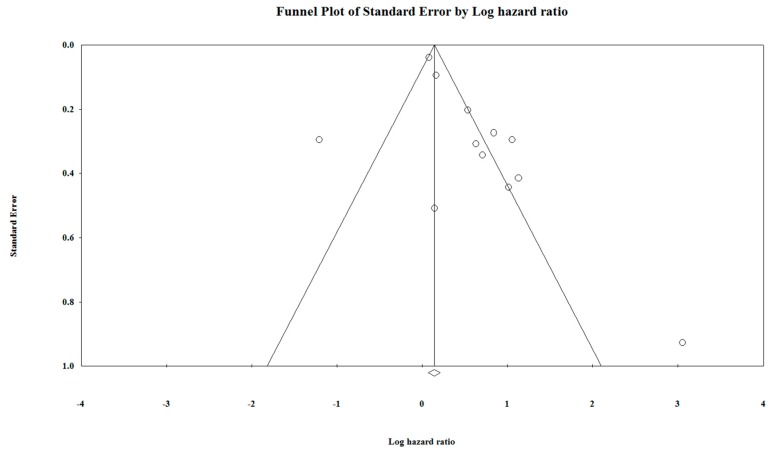
Funnel plot of studies included in the meta-analysis.

**Figure 7 cancers-11-00900-f007:**
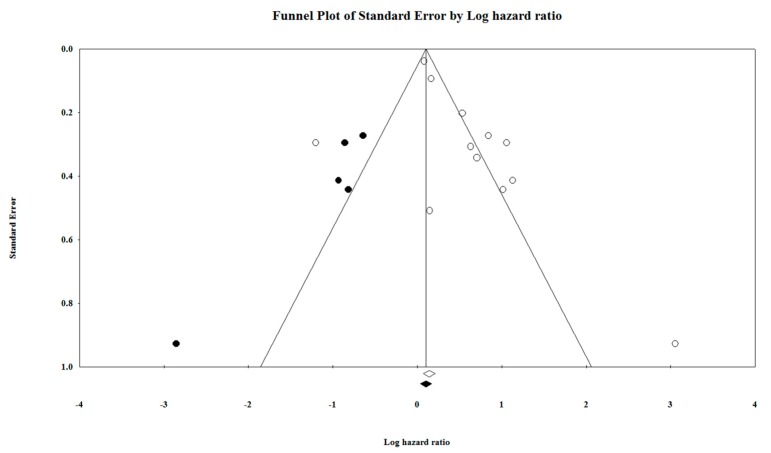
Funnel plot of studies including the imputed studies in our meta-analysis.

**Table 1 cancers-11-00900-t001:** Characteristics of 43 included studies.

S.No	Author	Ethnicity	Period of Study	Drug	No. of Samples (Cancer/Normal)	Cell Culture Type	Resistant Cells	miRNA	miRNA Profiling Platform	Pathways Associated with/Gene
1	Yang, R. et al. (2017) [55]	Chinese	2013–2016	GEM	157/157	Human PC cell lines Capan-2, HPAC, SW1990, PANC-1, CFPAC-1, BXPC-3, ASPC-1, PATU-8988, HPDE6-C7 and HPNE	PANC-1-GR and BXPC-3 GR	3656	Taqman microRNA Reverse Transcription kits (Thermo Fisher Scientific, Dreieich, Germany)	Ras Homolog Family Member F (RHOF)/Epithelial–mesenchymal transition (EMT)
2	Xiao, G. et al. (2017) [56]	Chinese	2015–2016	GEM	15/15	HPDE6-C7, PC cell lines Panc-1 and BxPc3	NM	Let-7a	TAKARA PrimeScript Kit	C-X-C chemokine receptor type 4 (CXCR4)/let-7a/High-mobility group AT-hook 2 (HMGA2)
3	Hiramoto, H. et al. (2017) [57]	Japanese	2000–2011	GEM	50	Panc1, KP4-4, SU.86.86, BxPC3 and MDA-MB-231	NM	509-5p, 1243	Custom Taqman miRNA Assays kit (Applied Biosystems, San Diego, CA, USA)	E-cadherin
4	Chaudhary, A.K. et al. (2017) [58]	American	NM	GEM	NM	HPDE	GEM-resistant MIA PaCa-2R cells	205-5p	SYBR Green-based pathway-focused miScript miRNA PCR Array (catalog number 102ZF, Qiagen, MD, USA) using Roche Light Cycler 480^®^ (Roche, Indianapolis, IN, USA)	K-ras, Caveolin-1, and Ki-67
5	Liu. F. et al. (2017) [59]	Chinese	January 2010–December 2014	GEM	87	BxPC-3, Panc-1, Capan-2, SW1990, Paca-2, AsPc-1, and CFPAC-1, HEK293T and HPDE	Capan-2, Panc-1, and AsPc-1	153	SYBR Premix Ex Taq (TaKaRa, Dalian, China) and run with an Applied Biosystems ViiATM 7 Real-Time PCR System (Applied Biosystems)	Snail
6	Mikamori, M. et al. (2017) [60]	Japanese	March 2007–August 2015	GEM	45	Panc1, MiaPaCa2, and PSN1 cell lines	Panc1-GR1, -GR3, and -GR4 cells	155	TaqMan MicroRNA Assays (Applied Biosystems) and the ABI7900HT system (Applied Biosystems)	Anti-apoptotic (RAB27B)
7	Hu, H. et al. (2016) [61]	Chinese	NM	GEM	15/15	PANC-1	NM	101	TaqMan microRNA assay using ABI-7300 Real-Time machine (Shanghai, China)	DNA-dependent protein kinase catalytic subunit (DNA-PKcs)
8	Amponsah, P. et al. (2016) [62]	Deutsch	NM	GEM	92/5	ASAN-PaCa, BxPC-3, AsPC-1 and MIA-PaCa2	Bx-GEM	210	Human HT-12 v4 Expression Bead Chip Kit or the Human miR Microarray (Release 19.0)	ABCC5
9	Li, C. et al. (2016) [63]	Chinese	2013-2015	GEM	31/31	HPDE6, PANC-1, MIAPaCa-2 and SW1990 cells	NM	124	TaqMan microRNA assays (Applied Biosystems)	miR-124/polypyrimidine tract binding protein 1 (PTBP1)/Pyruvate kinase (PKM2)
10	Li, J. et al. (2016) [64]	Chinese	NM	GEM	84/20	HPC-Y5, AsPC-1, PANC-1, BxPC-3, Hs766t and CFPAC-1	NM	506	Agilent Array	Sphingosine kinase 1 (SPHK1)/Protein kinase B(Akt)/nuclear factor kappa-light-chain-enhancer of activated B cells (NF-κB)
11	Gu, J. et al. (2016) [65]	Chinese	2008–2010	GEM	58	PanC-1, Mia Paca-2 and HEK-293T	NM	17-5p	SYBR Green Mix (Roche) using All-in-One miRNA qPCR Detection Kit (GeneCopoeia, Rock, MD, USA)	Phosphatase and tensin homolog (PTEN)
12	Tian, X. et al. (2016) [66]	American	March 2009–September 2013	GEM, Lapatinib, and Capecitabine	17	PANC-1, MIA PaCa-2 and BXCP-3 cell lines	NM	7, 21, 210, 221	RT2 miRNA first strand kit (Qiagen, Inc.) and Applied Biosystems 7900HT Fast Real-Time PCR system (Thermo Fisher Scientifc, Inc., Waltham, MA, USA)	Epidermal growth factor receptor (EGFR)1 and human epidermal growth factor receptor (HER)2 pathways
13	Fan, P. et al. (2016) [67]	Deutsch	NM	GEM	21	ASAN-PaCa, AsPC-1, PANC-1, MIA-PaCa2 and BxPC-3	Bx-GEM	101-3p	Human HT-12 v4 Expression Bead Chip Kit or the Human miR Microarray (Release 19.0).	Ribonucleotide reductase M1 (RRM1)
14	Yao, J. et al. (2016) [68]	Chinese	NM	GEM	26	SW1990 and HEK293T cells	SW1990GZ	125a	TaqMan MicroRNA Reverse Transcription Kit (Takara), miRscript SYBR Green PCR Kit and SYBR Green PCR Kit (Takara, Dalian, China)	TNF Alpha-Induced Protein 3 (A20)
15	Ren, Z. et al. (2016) [69]	Chinese	NM	GEM	10/10	L3.6pl, BxPC-3, CFPAC, MiaPaCa-2, ASPC-1, PANC-1, MPanc96, HPAC, SU86.86 and HS766T	NM	203	mirVana RT-qPCR miRNA Detection kit (cat no. AM7659; Ambion, Austin, TX, USA)	Salt-inducible kinase 1 (SIK1)
16	Chen, M. et al. (2015) [70]	Chinese	2008–2011	GEM	124/10	PANC-1 and BXPC3	NM	181c	miRNA-specific TaqMan MiRNA Assay Kit (Applied Biosystems).	Mammalian STE20-like protein kinase 1/2 (MST1/2), and large tumour suppressor 1/2 (LATS1/2), together with the adaptor proteins Salvador homolog 1 (SAV1) and MOB kinase activator 1 (MOB1) (Hippo signalling pathway)
17	Miyamae, M. et al. (2015) [71]	Japanese	January 2010–April 2013	GEM	94/68	PK-45H, PANC-1, PK-59, KP4-1, and PK-1	NM	550a, 557, 575, 615-5p, 675, 744	3D-Gene miRNA microarray platform (Toray Industries, Kamakura, Japan and human TaqMan MicroRNA Assay Kit (Applied Biosystems, Foster City, CA, USA)	NM
18	Zhang, W. et al. (2015) [72]	Chinese	NM	GEM	19	HPAC, BxPC-3, Colo357, and L3.6pl	ASPC-1, Panc-1 and MiaPaCa-2	15b, 155, 212	mirVana qRT-PCR miRNA detection kit (Ambion)	SMAD specific E3 ubiquitin protein ligase 2 (SMURF2)
19	Yu, C. et al. (2015) [73]	Chinese	2013–2014	5-FU	18	AsPC-1, BxPc-3, Capan-1, Capan-2, CFPAC-1, PANC-1, MIA PaCa-2 & SW1990	NM	138-5p	Fluorescence-activated cell sorting (FACSnCanto II flow cytometer; BD Biosciences, San Jose, CA, USA)	Vimentin (VIM)
20	Liang, C. et al. (2015) [74]	Chinese	2010–2012	GEM	106	PCI35 & PCI55, SW1990, MiaPaca-2, PANC-1, BxPC-3, Capan-1	NM	33a	NM	AKT/Gsk-3β/β-catenin pathway
21	Liu, Y. et al. (2015) [75]	Chinese	2007–2010	5-FU, GEM	86/41	AsPC-1, BXPC-3, SW1990, MIAPaCa-2, PANC-1 & HPDE	NM	494	NM	miR-494/c-Myc/SIRT1 pathway
22	Meidhof, S. et al. (2015) [76]	Deutsch	NM	GEM	27/27	Panc-1, MDA-MB-231, BxPC3, H358, DU-145, hPaca-1 and hPaca-2	BxPC3 GEM-resistant cells, Tarceva-resistant H358 cells	203	Roche LightCycler 480	ZEB-1
23	Zhao, Y. et al. (2015) [77]	Deutsch	NM	GEM	28/28	L3.6pl	L3.6pl - GemR	21,221	miScript SYBR^®^ Green PCR Kit (Qiagen, USA)	NM
24	Li, Z. et al. (2014) [78]	Chinese	2013-2014	GEM	23/23	AsPC1, BxPc-3, Capan-1, Capan-2, CFPAC-1, PANC-1, MIA PaCa-2, SW1990	NM	100	TaqMan miRNA Assay (Applied Biosystems)	FGFR3
25	Xu, J. et al. (2014) [79]	Chinese	NM	GEM	87	SW1990, MiaPaCa-2	SW1990/GEM	497	NM	FGF/FGFR signalling pathway
26	Hasegawa, S. et al. (2014) [80]	Japanese	2007-2010	5-FU, GEM	24	Panc1-P, Panc1-GR	Panc-1GemR	1246	Comparative CT method	CCNG2
27	Lai, I.-L. et al. (2014) [81]	Americans	NM	GEM	NM	Panc-1, AsPC-1 and BxPC-3	Panc-1GemR, BxPC3GemR and AsPC-1GemR	520f	Bio-Rad CFX Manager 2.1 detection system and miScript PCR starter kit (Qiagen)	ATM/ATR checkpoint pathway
28	Song, W.-F. et al. (2013) [82]	Chinese	2010–2012	GEM	41	BxPc3, HPAF, HPAC, Capan, PANC-1 and PL-45 cell lines	HPAC and PANC-1/GEM	21	Specific Taqman MicroRNA assays (Applied Biosystems)	PTEN/Akt pathway
29	Peng, F. et al. (2013) [83]	Chinese	2010–2011	5-FU	14	TFK-1, QBC939 cell line	NM	220b, 200c and 429	mirVana miRNA Isolation Kit (Ambion, Austin, TX, USA), Agilent Human miRNA Microarray Kit (V2) (Agilent Inc, Santa Clara, CA, USA) for analysis.	SUZ12, ROCK2 direct targets
30	Nagano, H. et al. (2013) [84]	Japanese	September 1999–February 2004	GEM	18	MIAPaCa-2, PSN-1, BxPC-3, Panc-1	NM	29a	TRIzol agent (Invitrogen, Carlsbad, CA, USA)	Wnt/β-catenin signaling pathway
31	Wei, F. et al. (2013) [85]	Chinese	NM	Radiation and AZD8055	NM	PANC-1, Capan-2, BxPC-3	NM	99b	NM	mTOR
32	Iwagami, Y. et al. (2013) [86]	Japanese	1992–2008	GEM	66	MiaPaCa2 and PSN1	MiaPaCa2-RGs, PSN1-RGs	320c	NanoDrop ND-1000 spectrophotometer (NanoDrop Technologies, Wilmington, DE, USA	SMARCC1 mediated anti-cancer effect of GEM
33	Bhutia, Y.D. et al. (2013) [87]	Americans	NM	GEM	10/2	MIA PaCa-2	L3.6pl and Capan-1/GEM	let-7a	miRNA Isolation Kit and the TaqManH MicroRNA Reverse Transcription Kit (Applied Biosystems)	RRM2
34	Wang, P. et al. (2012) [88]	Chinese	Cohort1: 2003–2005 Cohort2: 2009–2010	GEM	NM	Panc-1, BxPC3	NM	21	RecoverAll Total Nucleic Acid Isolation Kit (Ambion)	FasL/Fas pathway
35	Singh, S. et al. (2013) [89]	NM	NM	GEM	NM	MIA PaCa-2	MI PaCa-2/GEM	7, 146, 205	SYBR Green dye universal master mix on a Light Cycler 480 (Roche, Indianapolis)	Class III b-tubulin (TUBB3)
36	Brabletz, S. et al. (2011) [90]	NM	NM	GSI	NM	Panc1, HPAF2, MCF7, MiaPaCa2	NM	200	Pfu Ultra Hotstart 2 Master Mix (Stratagene, Santa Clara)	Notch signaling
37	Ali, S. et al. (2010) [91]	NM	NM	GEM, OHP, tarceva	50/10	MIAPaCa-2, AsPc-1	MIAPaCa-GR (GEM resistant), AsPc-1OR (oxaliplatin resistant), MIAPaCa-GTR, AsPc-1GTR (GEM and tarceva resistant)	21, 146a, 200b, 200c, 221, let-7b and let-7d	TaqMan MicroRNA Assay kit (Applied Biosystems)	NM
38	Hwang, J.-H. et al. (2010) [92]	Korean, Italian	1999–2007 and 2001–2004	GEM, 5-FU	127	BxPc3, HPAF-II, HPAC, PANC1, PL45	NM	21	TaqMan-microRNA assays and the 7900 HT-Fast RealTime PCR (Applied Biosystems, Foster City, CA, USA)	NM
39	Giovannetti, E. et al. (2010) [93]	Deutsch	2001–2004	GEM	81	hTERT-HPNE, Hhs27, LPc006, LPc033, LPc067, LPc111, LPc167, PP437	NM	21	7500HT sequence detection system (Applied Biosystems)	PTEN and PI3K-Akt pathway
40	Moriyama, T. et al. (2009) [94]	Japanese	2000–2008	GEM	25/25	AsPC-1, KP-1N, KP-2, KP-3, PANC-1, SUIT-2 MIA PaCa-2, CAPAN-1, CAPAN-2, CFPAC-1, H48N, HS766T, SW1990, NOR-P1	NM	21	mirVana qRT-PCR miRNA Detection Kit, and mirVana Primer Sets (all from Ambion)	VEGF and MMP-2 and MMP-9
41	Xiong G et al. (2018) [95]	Chinese	NM	GEM	90/90	AsPC-1, BxPC-3, MiaPaCa-2, PANC-1, Su86. 86, T3M4	AsPC-1-Gem	10a-5p	Genepharma (Shanghai, China)	TFPA2C
42	Sun, D. et al. (2018)	Chinese	January 2007–December 2015	GEM	87/8	BxPC-3, PANC-1, AsPC-1, SW1990, Capan-1, Capan-2, CFPAC-1 and MIA PaCa-2	NM	374b-5p	LightCycler^®^ 480 SYBR-Green I Master (Roche Diagnostics, Basel, Switzerland).	BIAPRC-3 and XIAP
43	You, L. et al. (2018)	Chinese	NM	GEM	10	293T, MIA PaCa-2, Su.86.86, Capan-1, PANC-1, SW1990, BxPC-3 and AsPC-1	GEM-R cells BxPC-3 and PANC-1	1207	(Bio-Rad, Hercules, CA, USA)	PVT1

NM-Not Mentioned.

**Table 2 cancers-11-00900-t002:** miRNAs that are involved in chemoresistance and pathways that are modulated.

Chemoresistance			
Downregulated		Upregulated	
miRNA	Pathway	miRNA	Pathway
**GEM**		**GEM**	
210	ABCC5	17-5p	PTEN
124	PTBP1/PKM2	221	EGFR and HER pathways
101-3p	RRM1	203	SIK1
100	FGFR3	181c	MST1/2, and LATS1/2, together with the adaptor proteins SAV1 and MOB1 (Hippo signalling pathway)
497	FGF/FGFR signalling pathway	15b	SMURF2
7	TUBB3	21	NM
205	TUBB3	221	NM
374b-5p	BIAPRC-3 and XIAP	1246	CCNG2
**5-FU**		21	PTEN/Akt pathway
200c	SUZ12, ROCK2 direct targets	320c	SMARCC1 mediated the anti-cancer effect of GEM
220b	SUZ12, ROCK2 direct targets	155	Anti-apoptotic (RAB27B)
		21	NM
		221	NM
		21	VEGF and MMP-2 and MMP-9
		10a-5p	TFAP2C
		**5-FU**	
		1246	CCNG2

**Table 3 cancers-11-00900-t003:** miRNAs that are involved in chemosensitivity and pathways that are modulated.

Chemosensitivity			
Downregulated		Upregulated	
miRNA	Pathway	miRNA	Pathway
**GEM**		**GEM**	
3656	RHOF/EMT	509-5p	E-cadherin
Let-7a	CXCR4/HMGA2	1243	E-cadherin
205-5p	K-ras, Caveolin-1 and Ki-67	33a	AKT/Gsk-3β/β-catenin pathway
153	SNAIL	21	FasL/Fas pathway
101	DNA-PKcs	1207	PVT1
506	SPHK1/Akt/NF-κB	**5-FU**	
494	c-Myc/SIRT1 pathway	138-5p	vimentin
203	ZEB-1		
**5-FU**			
494	c-Myc/SIRT1 pathway		

**Table 4 cancers-11-00900-t004:** Publication bias of the included studies.

**Groups**	**Clinical Outcomes**	**Classic Fail-Safe N**		**Orwin Fail-Safe N**	**Begg and Mazumdar**			**Egger′s Regression**			**Dual and Tweedie (Random Effects)**				
		***z*-Value**	***p*-Value**	**HR in Observed**	**Tau**	***z*-Value**	***p*-Value**	**Intercept**	***p*-Value**	**df**	**Observed**	***q*-Value**	**Adjusted**	***q*-Value**	
**Main**	**Main Meta-analysis**	6.264	0	1.150	0.257	1.165	0.243	1.778	0.052	10	1.603	68.041	1.132	107.980	
				**Fixed**			**Mixed/Random**			**Hypothesis Test**					
**Subgroups**	**Heterogeneity**			**HR**	**95% CI**		**HR**	**95% CI**		**Fixed effects model**			**Random effects model**		
	**Q**	**P**	**I^2^**		**Low**	**High**		**Low**	**High**	**Z**	**P**	**Studies**	**Z**	**P**	**Studies**
miR-21	1.683	0.195	40.567	1.913	1.340	2.732	2.061	1.195	3.556	3.569	0	2	2.599	0.009	2

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
