# Peer review of "miRNA Predictors of Pancreatic Cancer Chemotherapeutic Response: A Systematic Review and Meta-Analysis"

_cancers, 2019, doi:10.3390/cancers11070900_

Reviewer 1 Report

The review manuscript by Madhav and collaborators is a vast and serious systematic analysis of the past ten years of publications about miRNA field associated with pancreatic cancer parameters. This is a sound work which will be helpful for scientists involved in pancreatic cancer research.

I have only minor comments and suggestion that could improve the quality and accessibility to most people.

- Most figures need to be improved at the form level. Missing characters (such as in Fig. 2) or too small (such as in fig. 4) need to be improved. Fig.3 is a nice way to show the data but, here as well, the shape needs to be improved as this look like a draft.

- There are a lot of statistical analysis which is rather good. However, it will be easier for profane people if some brief explanations are given, as the authors did sometime in the manuscript. For example, in the section 5, authors should explain what is a value of 0.16667 for the Begg and Mazumdar test, or what means a value of 1.24762 at the Egger’s test of the Intercept. Another example, in section 4, what means a value of 91.23 for heterogeneity.

- Some sentences could be improved. For example, in the introduction section “With regards to India, cancer is one of the leading causing of death in India behind cardiovascular disease” should be replace by “With regards to India, cancer is one of the leading causing of death behind cardiovascular disease”. Another example, “According to an  estimate, the estimated annual PC burden in India in 2001 was 14,230 cases.” should be replaced by “According to a study, the estimated annual PC burden in India in 2001 was 14,230 cases.” or any equivalent.

Author Response

To

Dr. Jochen Sven Utikal

Guest Editor

Dr Jochen Sven Utikal

We would like to thank once again the Cancers (MDPI) Journal’s Editorial and Reviewers team for reviewing our manuscript and providing valuable comments. Our team firmly believes that the comments and feedback from your esteemed reviewers were constructive and would enhance the quality of the paper. Please find the revised manuscript addressing issues identified by editors and reviewers team in considering the above review paper for publication.

We have amended the changes and highlighted the sections in the manuscript with track changes.

Reviewer 1:

Our sincere thanks for your valuable and thoughtful review. Our clinical oncology research team is pleased that your suggestions would improve the quality of our manuscript. It is greatly appreciated.

1.      Most figures need to be improved at the form level. Missing characters (such as in Fig. 2) or too small (such as in fig. 4) need to be improved. Fig.3 is a nice way to show the data but, here as well, the shape needs to be improved as this look like a draft.

We have modified the figures and improved the quality of the resolution. (Line no: 290-293; 371-373)

Figure 2. commonly performed in vitro assays in the included articles.

ISH: In-Situ Hybridization; IHC: Immuno Histo-Chemistry; TUNEL: Terminal deoxynucleotidyl transferase (TdT) dUTP Nick-End Labeling.

            Random effect model; I squared=83.833%; tau=0.403; Q value= 68.042; df=11

Figure 4. Forest plot for the HR and 95% CI of PC studies using meta-analysis.

2.      In the section 5, authors should explain what is a value of 0.16667 for the Begg and Mazumdar test, or what means a value of 1.24762 at the Egger’s test of the Intercept. Another example, in section 4, what means a value of 91.23 for heterogeneity.

As per the reviewer suggestion, we have tried to explain wherever possible. Big studies tend to be included in the meta-analysis regardless of their prognostic effect. Given this circumstances, there will be an inverse correlation between study size and effect size. Begg and Mazumdar rank order correlation was computed (Kendall's tau b) between the intervention effect and the standard error (which is driven primarily by sample size).

This value compares the effect size and variance with the tau value and the value closest to 1 correlate with greater publication bias (Line 401-402).

Both Classic and the Orwin fail-safe N inform the likelihood that studies are absent from the meta-analysis and that these studies, if included in the analysis, would shift the estimated effect size of HR of value toward the null.

Duval and Tweedie's Trim and Fill test sanctions imputing the missing studies that are likely to fall, add them to the analysis, and then recompute the combined effects of miRNAs.

3.      Some sentences could be improved. For example, in the introduction section “With regards to India, cancer is one of the leading causing of death in India behind cardiovascular disease” should be replace by “With regards to India, cancer is one of the leading causing of death behind cardiovascular disease”. Another example, “According to an  estimate, the estimated annual PC burden in India in 2001 was 14,230 cases.” should be replaced by “According to a study, the estimated annual PC burden in India in 2001 was 14,230 cases.” or any equivalent.

As per the reviewer suggestion, we have rephrased the sentences. (Line no: 148-150)

Reviewer 2:

Our sincere thanks for your valuable and thoughtful review. Our clinical oncology research team is pleased that your suggestions would improve the quality of our manuscript. It is greatly appreciated.

4.      This manuscript claims to provide a consensus on access to cancer care innovations in countries with limited resources. However, in neither the narrative nor indeed in the methods/results does it succeed in achieving this.

As per the reviewer comment, we tried to provide an overview and status of the cancer care in limited-resource countries, but we could not find many published studies. Therefore, we have mentioned this in the limitations section

5.      The abstract should be thoroughly rewritten. It currently consists of a sequence of words and sentences that do not necessarily convey an understandable meaning. For instance the list of therapies is confusing. Similarly, it’s not clear why the authors have chosen some “representative” studies on miRNA in different cellular models.

As per the reviewer comment, we have tried to ameliorate and address this issue as per the reviewer’s instructions. The representative studies are chosen to highlight the link between miRNA de-regulation and chemotherapeutic responses in pancreatic cancer. (Line no: 99-130)

6.      The specific research question is not clear from the introduction.

As per the reviewer suggestion, we have modified the objective statements/research questions to make the purpose of this study clearer. (Line no: 188-201)

7.      The manuscript is full of small and large errors and strange expressions. For example: “Our study aims to clarify the relationship between miRNA expression and chemotherapeutic resistance or susceptibility in PC through a systematic review and meta-analysis of extant literature.”

As per the reviewer suggestion, we have modified the sentence. (Line no: 199-201)

8.      Other statements lack clear explanation (e.g., “A study by Zhu et al. observed that miRNA 27a and 451 were upregulated in multidrug-resistant cell lines (A2780DX5 and KB-V1) paralleling the overexpression of P-glycoprotein (P-gp) [36]. In this situation, the transfection of miRNA-451 in doxorubicin-resistant MCF-7 cell lines increased the sensitivity to doxorubicin by [37].

As per the reviewer suggestion, we have modified the sentence. (Line no: 174-175)

9.      Drug resistance is classified as intrinsic if it is present before treatment and acquired”

We totally agree with the reviewer in case of drug resistance present before the start of treatment to a particular drug it is considered as intrinsic while drug resistance that develops after a treatment is categorised as acquired.

10.  The description of the methods is very poor. One cannot simply report the inclusions criteria as a list. The Authors should clarify how, by what criteria, through what process? There are a suite of methods available for this sort of analysis.

We have reported the inclusion criteria according to the Preferred Reporting Items for Systematic Review and Meta-analysis (PRISMA) guidelines where each criteria is being designed to optimise the quality of selected and included studies in order to minimising possible selection bias. The selection process for the criteria has been double-checked by our systematic review and meta-analysis team. The PROSPERO (International prospective register of systematic reviews by the National Institute of Health Research; https://www.crd.york.ac.uk/prospero/) team has reviewed this study protocol including selection criteria. In addition, we have a clear search strategy using MeSH search terms to select the studies for our analysis

11.  I do not understand the Figure 3. The authors should explain how they selected the miRNAs that are downregulated and upregulated as well as their targets.

We have selected the miRNAs based on reports and findings of the included studies because they analysed the miRNA expression using one of the mentioned methods such as RT-PCR or microarray in comparison with the internal control.

12.  The results reported in table 4 and Figure 5 for the study of Hwang J et al. are wrong: as reported in the abstract in this study: “Cox proportional hazards model analysis in the subgroup of patients treated with adjuvant therapy (N = 52) showed that lower than median miR-21 expression was associated with a significantly lower hazard ratio (HR) for death (HR = 0.316; 95%CI = 0.166-0.600; P = 0.0004) and recurrence (HR = 0.521; 95%CI = 0.280-0.967; P = 0.04)”. So, the high expression of miRNA-21 was associated with HR>1, as reported in the study by Wang et al., and Giovannetti et al. (in the same Figure 3)

According to the Reviewers suggestion, we have removed Hwang et al study from our meta-analysis and subgroup analysis in addition to publication. We have amended them in our figures.

13.  In general the whole paper is biased by the fact that there is no critical analysis of the data listed. They are just reported and displayed.

·         As this study was designed based on PRISMA guidelines and also registered in PROSPERO (each guidelines and statements have their own criteria for reporting and presenting the results), the analysis needs to be heavily reliant on guidelines and statement reporting of aforementioned methods.

·         We had very strict selection criteria, search strategy and have performed different parameters of publication bias (Begg and Mazumdar rank order correlation, Classic and the Orwin fail-safe N and Duval and Tweedie's Trim and Fill test) in order to avoid any sort of bias in our study and analysis.

·         The report and display structure were adopted PRISMA checklist in order to minimise any possibility of author opinions influencing the results or its subsequent interpretations.

·         This should allow for ease of dissemination and use as a repository of collated information for future studies as well.

14.  The discussion is equally impenetrable, and many statements are simply not correlated with the data (for instance, why the authors wrote a paragraph on “In a study on lung cancer cells, miR-17-5p downregulation was associated with an increased 282 expression of beclin 1 gene which is an autophagy modulator in the survival pathway.”? The fact that “this miRNA belongs to the miR-17-92 cluster and is up-regulated in PC where it is linked 284 to pancreatic carcinogenesis.” Does not related to its potential role in chemoresistance)

·         We would like to highlight the molecular behaviour of that particular miRNA in different cancers, thus have compared our findings with lung cancer where miR-17-5p is downregulated, but in pancreatic cancer it is upregulated, and associated with the PTEN pathway and therefore, PI3K-Akt signalling.

·         We tried to say that the same miRNA when differentially regulated (up or down) switches its pathway in different cancers and can exhibit a discordant expression effects on carcinogenesis and theragnosis.

15.  The conclusions (i.e., “This knowledge could drive the choice of chemotherapeutic regimens used in patients”) should be rewritten.

We have modified the sentence.

Reviewer 3:

Our sincere thanks for your valuable and thoughtful review. Our clinical oncology research team is pleased that your suggestions would improve the quality of our manuscript. It is greatly appreciated.

Specific comments:

16.  In Tables 1 and 2 the acronym “NM” should be defined and/or spelled out.

We have mentioned in the foot notes. (Line no: 283)

17.  There is no “call-out” to Figure 3 in the text; an improved figure legend for Figure 3 would help clarify the information presented in this figure. It is not clear what is meant by “hallmarks of pancreatic cancer.” Green and red sections denote miRNAs that are up-regulated and down-regulated, respectively; is this up- or down-regulation the basal expression of these miRNAs in pancreatic cancers compared to healthy tissue or is this in response to chemotherapy? Finally, do the gene names on the outside edge of the circle reference targets of the miRNAs or the pathways/mechanisms by which their expression is regulated in pancreatic cancer? A clearer figure legend should address and provide information to answer these questions.

We have modified the figure legends. The outside edge of the circle refers to the drug target by the miRNAs. (Line no: 310-312; 342)

18. A “call-out” to Table 2 in the text would be useful (likely inserted in paragraphs on page 15). A re-labelling of Table 2 that describes “chemosensitivity” to Table 3 would be useful, as the “chemoresistance” and “chemosensitivity” tables are indeed two separate tables (as of now both are labelled Table 2). The new Table 3 should be referenced in the text (likely in the paragraph on page 16).

We have renamed the table as 2 and 3, respectively, and listed out in the text section. (Line no: 344)

19.  An expanded comment in the discussion section on the meta-analysis of miR-21 involvement in pancreatic cancer would be useful; the authors’ analysis suggests that mIR-21 expression does not necessarily predict chemoresistance or patient survival (data in Figure 5): this is an important finding and should be discussed.

We would like to refrain from commenting heavily on miR-21 as the power of the analysis is too low to comment on the results obtained from the miR-21 subgroup. Although our results do show that miR-21 may not have any significant effect on directing patient survival or chemotherapeutic response, the fact that this conclusion was draw from a total of 2 studies precludes us from forming any concrete responses. Therefore, we have remained silent of the matter, until further assessment is conducted, and more data is available.

Many Thanks for your valuable time.

Thanking you,

Dr Jayaraj

Reviewer 2 Report

Although from an equity point of view the research undertaken by this group is certainly of interest, the article is poorly written and therefore extremely hard to understand. 

This manuscript claims to provide a consensus on access to cancer care innovations in countries with limited resources. However, in neither the narrative nor indeed in the methods/results does it succeed in achieving this.

The abstract should be thoroughly rewritten. It currently consists of a sequence of words and sentences that do not necessarily convey an understandable meaning. For instance the list of therapies is confusing. Similarly, it’s not clear why the authors have chosen some “representative” studies on miRNA in different cellular models.

The introduction is quite long and does not introduce the theme that is at stake in the article, namely research efforts on miRNA involved in pancreatic cancer resistance. The specific research question is not clear from the introduction.

The manuscript is full of small and large errors and strange expressions. For example: “Our study aims to clarify the relationship between miRNA expression and chemotherapeutic resistance or susceptibility in PC through a systematic review and meta-analysis of extant literature.”

Other statements lack clear explanation (e.g., “A study by Zhu et al. observed that miRNA 27a and 451 73 were upregulated in multidrug-resistant cell lines (A2780DX5 and KB-V1) paralleling the 74 overexpression of P-glycoprotein (P-gp) [36]. In this situation, the transfection of miRNA-451 in 75 doxorubicin-resistant MCF-7 cell lines increased the sensitivity to doxorubicin by [37]. 76

Drug resistance is classified as intrinsic if it is present before treatment and acquired”

The description of the methods is very poor. One cannot simply report the inclusions criteria as a list. The Authors should clarify how, by what criteria, through what process? There are a suite of methods available for this sort of analysis.

I do not understand the Figure 3. The authors should explain how they selected the miRNAs that are downregulated and upregulated as well as their targets.

The results reported in table 4 and Figure 5 for the study of Hwang J et al are wrong: as reported in the abstract in this study: “Cox proportional hazards model analysis in the subgroup of patients treated with adjuvant therapy (N = 52) showed that lower than median miR-21 expression was associated with a significantly lower hazard ratio (HR) for death (HR = 0.316; 95%CI = 0.166-0.600; P = 0.0004) and recurrence (HR = 0.521; 95%CI = 0.280-0.967; P = 0.04)”. So, the high expression of miRNA-21 was associated with HR>1, as reported in the study by Wang et al, and Giovannetti et al. (in the same Figure 3)

In general the whole paper is biased by the fact that there is no critical analysis of the data listed. They are just reported and displayed.

The discussion is equally impenetrable, and many statements are simply not correlated with the data (for instance, why the authors wrote a paragraph on “In a study on lung cancer cells, miR-17-5p downregulation was associated with an increased 282 expression of beclin 1 gene which is an autophagy modulator in the survival pathway.”? The fact that “this miRNA belongs to the miR-17-92 cluster and is up-regulated in PC where it is linked 284 to pancreatic carcinogenesis.” Does not related to its potential role in chemoresistance)

The conclusions (i.e., “This knowledge could drive the choice of chemotherapeutic regimens used in patients”) should be rewritten.

Author Response

To

Dr. Jochen Sven Utikal

Guest Editor

Dr Jochen Sven Utikal

We would like to thank once again the Cancers (MDPI) Journal’s Editorial and Reviewers team for reviewing our manuscript and providing valuable comments. Our team firmly believes that the comments and feedback from your esteemed reviewers were constructive and would enhance the quality of the paper. Please find the revised manuscript addressing issues identified by editors and reviewers team in considering the above review paper for publication.

We have amended the changes and highlighted the sections in the manuscript with track changes.

Reviewer 1:

Our sincere thanks for your valuable and thoughtful review. Our clinical oncology research team is pleased that your suggestions would improve the quality of our manuscript. It is greatly appreciated.

1.      Most figures need to be improved at the form level. Missing characters (such as in Fig. 2) or too small (such as in fig. 4) need to be improved. Fig.3 is a nice way to show the data but, here as well, the shape needs to be improved as this look like a draft.

We have modified the figures and improved the quality of the resolution. (Line no: 290-293; 371-373)

Figure 2. commonly performed in vitro assays in the included articles.

ISH: In-Situ Hybridization; IHC: Immuno Histo-Chemistry; TUNEL: Terminal deoxynucleotidyl transferase (TdT) dUTP Nick-End Labeling.

            Random effect model; I squared=83.833%; tau=0.403; Q value= 68.042; df=11

Figure 4. Forest plot for the HR and 95% CI of PC studies using meta-analysis.

2.      In the section 5, authors should explain what is a value of 0.16667 for the Begg and Mazumdar test, or what means a value of 1.24762 at the Egger’s test of the Intercept. Another example, in section 4, what means a value of 91.23 for heterogeneity.

As per the reviewer suggestion, we have tried to explain wherever possible. Big studies tend to be included in the meta-analysis regardless of their prognostic effect. Given this circumstances, there will be an inverse correlation between study size and effect size. Begg and Mazumdar rank order correlation was computed (Kendall's tau b) between the intervention effect and the standard error (which is driven primarily by sample size).

This value compares the effect size and variance with the tau value and the value closest to 1 correlate with greater publication bias (Line 401-402).

Both Classic and the Orwin fail-safe N inform the likelihood that studies are absent from the meta-analysis and that these studies, if included in the analysis, would shift the estimated effect size of HR of value toward the null.

Duval and Tweedie's Trim and Fill test sanctions imputing the missing studies that are likely to fall, add them to the analysis, and then recompute the combined effects of miRNAs.

3.      Some sentences could be improved. For example, in the introduction section “With regards to India, cancer is one of the leading causing of death in India behind cardiovascular disease” should be replace by “With regards to India, cancer is one of the leading causing of death behind cardiovascular disease”. Another example, “According to an  estimate, the estimated annual PC burden in India in 2001 was 14,230 cases.” should be replaced by “According to a study, the estimated annual PC burden in India in 2001 was 14,230 cases.” or any equivalent.

As per the reviewer suggestion, we have rephrased the sentences. (Line no: 148-150)

Reviewer 2:

Our sincere thanks for your valuable and thoughtful review. Our clinical oncology research team is pleased that your suggestions would improve the quality of our manuscript. It is greatly appreciated.

4.      This manuscript claims to provide a consensus on access to cancer care innovations in countries with limited resources. However, in neither the narrative nor indeed in the methods/results does it succeed in achieving this.

As per the reviewer comment, we tried to provide an overview and status of the cancer care in limited-resource countries, but we could not find many published studies. Therefore, we have mentioned this in the limitations section.

5.      The abstract should be thoroughly rewritten. It currently consists of a sequence of words and sentences that do not necessarily convey an understandable meaning. For instance the list of therapies is confusing. Similarly, it’s not clear why the authors have chosen some “representative” studies on miRNA in different cellular models.

As per the reviewer comment, we have tried to ameliorate and address this issue as per the reviewer’s instructions. The representative studies are chosen to highlight the link between miRNA de-regulation and chemotherapeutic responses in pancreatic cancer. (Line no: 99-130)

6.      The specific research question is not clear from the introduction.

As per the reviewer suggestion, we have modified the objective statements/research questions to make the purpose of this study clearer. (Line no: 188-201)

7.      The manuscript is full of small and large errors and strange expressions. For example: “Our study aims to clarify the relationship between miRNA expression and chemotherapeutic resistance or susceptibility in PC through a systematic review and meta-analysis of extant literature.”

As per the reviewer suggestion, we have modified the sentence. (Line no: 199-201)

8.      Other statements lack clear explanation (e.g., “A study by Zhu et al. observed that miRNA 27a and 451 were upregulated in multidrug-resistant cell lines (A2780DX5 and KB-V1) paralleling the overexpression of P-glycoprotein (P-gp) [36]. In this situation, the transfection of miRNA-451 in doxorubicin-resistant MCF-7 cell lines increased the sensitivity to doxorubicin by [37].

As per the reviewer suggestion, we have modified the sentence. (Line no: 174-175)

9.      Drug resistance is classified as intrinsic if it is present before treatment and acquired”

We totally agree with the reviewer in case of drug resistance present before the start of treatment to a particular drug it is considered as intrinsic while drug resistance that develops after a treatment is categorised as acquired.

10.  The description of the methods is very poor. One cannot simply report the inclusions criteria as a list. The Authors should clarify how, by what criteria, through what process? There are a suite of methods available for this sort of analysis.

We have reported the inclusion criteria according to the Preferred Reporting Items for Systematic Review and Meta-analysis (PRISMA) guidelines where each criteria is being designed to optimise the quality of selected and included studies in order to minimising possible selection bias. The selection process for the criteria has been double-checked by our systematic review and meta-analysis team. The PROSPERO (International prospective register of systematic reviews by the National Institute of Health Research; https://www.crd.york.ac.uk/prospero/) team has reviewed this study protocol including selection criteria. In addition, we have a clear search strategy using MeSH search terms to select the studies for our analysis

11.  I do not understand the Figure 3. The authors should explain how they selected the miRNAs that are downregulated and upregulated as well as their targets.

We have selected the miRNAs based on reports and findings of the included studies because they analysed the miRNA expression using one of the mentioned methods such as RT-PCR or microarray in comparison with the internal control.

12.  The results reported in table 4 and Figure 5 for the study of Hwang J et al. are wrong: as reported in the abstract in this study: “Cox proportional hazards model analysis in the subgroup of patients treated with adjuvant therapy (N = 52) showed that lower than median miR-21 expression was associated with a significantly lower hazard ratio (HR) for death (HR = 0.316; 95%CI = 0.166-0.600; P = 0.0004) and recurrence (HR = 0.521; 95%CI = 0.280-0.967; P = 0.04)”. So, the high expression of miRNA-21 was associated with HR>1, as reported in the study by Wang et al., and Giovannetti et al. (in the same Figure 3)

According to the Reviewers suggestion, we have removed Hwang et al study from our meta-analysis and subgroup analysis in addition to publication. We have amended them in our figures.

13.  In general the whole paper is biased by the fact that there is no critical analysis of the data listed. They are just reported and displayed.

·         As this study was designed based on PRISMA guidelines and also registered in PROSPERO (each guidelines and statements have their own criteria for reporting and presenting the results), the analysis needs to be heavily reliant on guidelines and statement reporting of aforementioned methods.

·         We had very strict selection criteria, search strategy and have performed different parameters of publication bias (Begg and Mazumdar rank order correlation, Classic and the Orwin fail-safe N and Duval and Tweedie's Trim and Fill test) in order to avoid any sort of bias in our study and analysis.

·         The report and display structure were adopted PRISMA checklist in order to minimise any possibility of author opinions influencing the results or its subsequent interpretations.

·         This should allow for ease of dissemination and use as a repository of collated information for future studies as well.

14.  The discussion is equally impenetrable, and many statements are simply not correlated with the data (for instance, why the authors wrote a paragraph on “In a study on lung cancer cells, miR-17-5p downregulation was associated with an increased 282 expression of beclin 1 gene which is an autophagy modulator in the survival pathway.”? The fact that “this miRNA belongs to the miR-17-92 cluster and is up-regulated in PC where it is linked 284 to pancreatic carcinogenesis.” Does not related to its potential role in chemoresistance)

·         We would like to highlight the molecular behaviour of that particular miRNA in different cancers, thus have compared our findings with lung cancer where miR-17-5p is downregulated, but in pancreatic cancer it is upregulated, and associated with the PTEN pathway and therefore, PI3K-Akt signalling.

·         We tried to say that the same miRNA when differentially regulated (up or down) switches its pathway in different cancers and can exhibit a discordant expression effects on carcinogenesis and theragnosis.

15.  The conclusions (i.e., “This knowledge could drive the choice of chemotherapeutic regimens used in patients”) should be rewritten.

We have modified the sentence.

Reviewer 3:

Our sincere thanks for your valuable and thoughtful review. Our clinical oncology research team is pleased that your suggestions would improve the quality of our manuscript. It is greatly appreciated.

Specific comments:

16.  In Tables 1 and 2 the acronym “NM” should be defined and/or spelled out.

We have mentioned in the foot notes. (Line no: 283)

17.  There is no “call-out” to Figure 3 in the text; an improved figure legend for Figure 3 would help clarify the information presented in this figure. It is not clear what is meant by “hallmarks of pancreatic cancer.” Green and red sections denote miRNAs that are up-regulated and down-regulated, respectively; is this up- or down-regulation the basal expression of these miRNAs in pancreatic cancers compared to healthy tissue or is this in response to chemotherapy? Finally, do the gene names on the outside edge of the circle reference targets of the miRNAs or the pathways/mechanisms by which their expression is regulated in pancreatic cancer? A clearer figure legend should address and provide information to answer these questions.

We have modified the figure legends. The outside edge of the circle refers to the drug target by the miRNAs. (Line no: 310-312; 342)

18. A “call-out” to Table 2 in the text would be useful (likely inserted in paragraphs on page 15). A re-labelling of Table 2 that describes “chemosensitivity” to Table 3 would be useful, as the “chemoresistance” and “chemosensitivity” tables are indeed two separate tables (as of now both are labelled Table 2). The new Table 3 should be referenced in the text (likely in the paragraph on page 16).

We have renamed the table as 2 and 3, respectively, and listed out in the text section. (Line no: 344)

19.  An expanded comment in the discussion section on the meta-analysis of miR-21 involvement in pancreatic cancer would be useful; the authors’ analysis suggests that mIR-21 expression does not necessarily predict chemoresistance or patient survival (data in Figure 5): this is an important finding and should be discussed.

We would like to refrain from commenting heavily on miR-21 as the power of the analysis is too low to comment on the results obtained from the miR-21 subgroup. Although our results do show that miR-21 may not have any significant effect on directing patient survival or chemotherapeutic response, the fact that this conclusion was draw from a total of 2 studies precludes us from forming any concrete responses. Therefore, we have remained silent of the matter, until further assessment is conducted, and more data is available.

Many Thanks for your valuable time.

Thanking you,

Dr Jayaraj

Reviewer 3 Report

In their manuscript entitled “miRNA predictors of pancreatic cancer chemotherapeutic response: a systematic review and meta-analysis” Madhav et al. provide a comprehensive review of the past 10 years of literature describing  the correlations between miRNA and chemotherapeutic response and patient survival for pancreatic cancer. The authors have done a good job at presenting information on the relevant literature and studies included in their analyses, as well as information on how these data were analyzed and the interpretations of the results. This review and meta-analysis is a valuable document for investigators interested in the role of miRNA expression in pancreatic cancer, particularly with reference to the correlation of miRNA expression with therapeutic response and patient survival.

Specific comments:

1.      In Tables 1 and 2 the acronym “NM” should be defined and/or spelled out.

2.      There is no “call-out” to Figure 3 in the text; an improved figure legend for Figure 3 would help clarify the information presented in this figure. It is not clear what is meant by “hallmarks of pancreatic cancer.” Green and red sections denote miRNAs that are up-regulated and down-regulated, respectively; is this up- or down-regulation the basal expression of these miRNAs in pancreatic cancers compared to healthy tissue or is this in response to chemotherapy? Finally, do the gene names on the outside edge of the circle reference targets of the miRNAs or the pathways/mechanisms by which their expression is regulated in pancreatic cancer? A clearer figure legend should address and provide information to answer these questions.

3.      A “call-out” to Table 2 in the text would be useful (likely inserted in paragraphs on page 15). A re-labelling of Table 2 that describes “chemosensitivity” to Table 3 would be useful, as the “chemoresistance” and “chemosensitivity” tables are indeed two separate tables (as of now both are labelled Table 2). The new Table 3 should be referenced in the text (likely in the paragraph on page 16).

4.      An expanded comment in the discussion section on the meta-analysis of miR-21 involvement in pancreatic cancer would be useful; the authors’ analysis suggests that mIR-21 expression does not necessarily predict chemoresistance or patient survival (data in Figure 5): this is an important finding and should be discussed.

Author Response

To

Dr. Jochen Sven Utikal

Guest Editor

Dr Jochen Sven Utikal

We would like to thank once again the Cancers (MDPI) Journal’s Editorial and Reviewers team for reviewing our manuscript and providing valuable comments. Our team firmly believes that the comments and feedback from your esteemed reviewers were constructive and would enhance the quality of the paper. Please find the revised manuscript addressing issues identified by editors and reviewers team in considering the above review paper for publication.

We have amended the changes and highlighted the sections in the manuscript with track changes.

Reviewer 1:

Our sincere thanks for your valuable and thoughtful review. Our clinical oncology research team is pleased that your suggestions would improve the quality of our manuscript. It is greatly appreciated.

1.      Most figures need to be improved at the form level. Missing characters (such as in Fig. 2) or too small (such as in fig. 4) need to be improved. Fig.3 is a nice way to show the data but, here as well, the shape needs to be improved as this look like a draft.

We have modified the figures and improved the quality of the resolution. (Line no: 290-293; 371-373)

Figure 2. commonly performed in vitro assays in the included articles.

ISH: In-Situ Hybridization; IHC: Immuno Histo-Chemistry; TUNEL: Terminal deoxynucleotidyl transferase (TdT) dUTP Nick-End Labeling.

            Random effect model; I squared=83.833%; tau=0.403; Q value= 68.042; df=11

Figure 4. Forest plot for the HR and 95% CI of PC studies using meta-analysis.

2.      In the section 5, authors should explain what is a value of 0.16667 for the Begg and Mazumdar test, or what means a value of 1.24762 at the Egger’s test of the Intercept. Another example, in section 4, what means a value of 91.23 for heterogeneity.

As per the reviewer suggestion, we have tried to explain wherever possible. Big studies tend to be included in the meta-analysis regardless of their prognostic effect. Given this circumstances, there will be an inverse correlation between study size and effect size. Begg and Mazumdar rank order correlation was computed (Kendall's tau b) between the intervention effect and the standard error (which is driven primarily by sample size).

This value compares the effect size and variance with the tau value and the value closest to 1 correlate with greater publication bias (Line 401-402).

Both Classic and the Orwin fail-safe N inform the likelihood that studies are absent from the meta-analysis and that these studies, if included in the analysis, would shift the estimated effect size of HR of value toward the null.

Duval and Tweedie's Trim and Fill test sanctions imputing the missing studies that are likely to fall, add them to the analysis, and then recompute the combined effects of miRNAs.

3.      Some sentences could be improved. For example, in the introduction section “With regards to India, cancer is one of the leading causing of death in India behind cardiovascular disease” should be replace by “With regards to India, cancer is one of the leading causing of death behind cardiovascular disease”. Another example, “According to an  estimate, the estimated annual PC burden in India in 2001 was 14,230 cases.” should be replaced by “According to a study, the estimated annual PC burden in India in 2001 was 14,230 cases.” or any equivalent.

As per the reviewer suggestion, we have rephrased the sentences. (Line no: 148-150)

Reviewer 2:

Our sincere thanks for your valuable and thoughtful review. Our clinical oncology research team is pleased that your suggestions would improve the quality of our manuscript. It is greatly appreciated.

4.      This manuscript claims to provide a consensus on access to cancer care innovations in countries with limited resources. However, in neither the narrative nor indeed in the methods/results does it succeed in achieving this.

As per the reviewer comment, we tried to provide an overview and status of the cancer care in limited-resource countries, but we could not find many published studies. Therefore, we have mentioned this in the limitations section.

5.      The abstract should be thoroughly rewritten. It currently consists of a sequence of words and sentences that do not necessarily convey an understandable meaning. For instance the list of therapies is confusing. Similarly, it’s not clear why the authors have chosen some “representative” studies on miRNA in different cellular models.

As per the reviewer comment, we have tried to ameliorate and address this issue as per the reviewer’s instructions. The representative studies are chosen to highlight the link between miRNA de-regulation and chemotherapeutic responses in pancreatic cancer. (Line no: 99-130)

6.      The specific research question is not clear from the introduction.

As per the reviewer suggestion, we have modified the objective statements/research questions to make the purpose of this study clearer. (Line no: 188-201)

7.      The manuscript is full of small and large errors and strange expressions. For example: “Our study aims to clarify the relationship between miRNA expression and chemotherapeutic resistance or susceptibility in PC through a systematic review and meta-analysis of extant literature.”

As per the reviewer suggestion, we have modified the sentence. (Line no: 199-201)

8.      Other statements lack clear explanation (e.g., “A study by Zhu et al. observed that miRNA 27a and 451 were upregulated in multidrug-resistant cell lines (A2780DX5 and KB-V1) paralleling the overexpression of P-glycoprotein (P-gp) [36]. In this situation, the transfection of miRNA-451 in doxorubicin-resistant MCF-7 cell lines increased the sensitivity to doxorubicin by [37].

As per the reviewer suggestion, we have modified the sentence. (Line no: 174-175).

9.      Drug resistance is classified as intrinsic if it is present before treatment and acquired”

We totally agree with the reviewer in case of drug resistance present before the start of treatment to a particular drug it is considered as intrinsic while drug resistance that develops after a treatment is categorised as acquired.

10.  The description of the methods is very poor. One cannot simply report the inclusions criteria as a list. The Authors should clarify how, by what criteria, through what process? There are a suite of methods available for this sort of analysis.

We have reported the inclusion criteria according to the Preferred Reporting Items for Systematic Review and Meta-analysis (PRISMA) guidelines where each criteria is being designed to optimise the quality of selected and included studies in order to minimising possible selection bias. The selection process for the criteria has been double-checked by our systematic review and meta-analysis team. The PROSPERO (International prospective register of systematic reviews by the National Institute of Health Research; https://www.crd.york.ac.uk/prospero/) team has reviewed this study protocol including selection criteria. In addition, we have a clear search strategy using MeSH search terms to select the studies for our analysis.

11.  I do not understand the Figure 3. The authors should explain how they selected the miRNAs that are downregulated and upregulated as well as their targets.

We have selected the miRNAs based on reports and findings of the included studies because they analysed the miRNA expression using one of the mentioned methods such as RT-PCR or microarray in comparison with the internal control.

12.  The results reported in table 4 and Figure 5 for the study of Hwang J et al. are wrong: as reported in the abstract in this study: “Cox proportional hazards model analysis in the subgroup of patients treated with adjuvant therapy (N = 52) showed that lower than median miR-21 expression was associated with a significantly lower hazard ratio (HR) for death (HR = 0.316; 95%CI = 0.166-0.600; P = 0.0004) and recurrence (HR = 0.521; 95%CI = 0.280-0.967; P = 0.04)”. So, the high expression of miRNA-21 was associated with HR>1, as reported in the study by Wang et al., and Giovannetti et al. (in the same Figure 3)

According to the Reviewers suggestion, we have removed Hwang et al study from our meta-analysis and subgroup analysis in addition to publication. We have amended them in our figures.

13.  In general the whole paper is biased by the fact that there is no critical analysis of the data listed. They are just reported and displayed.

·         As this study was designed based on PRISMA guidelines and also registered in PROSPERO (each guidelines and statements have their own criteria for reporting and presenting the results), the analysis needs to be heavily reliant on guidelines and statement reporting of aforementioned methods.

·         We had very strict selection criteria, search strategy and have performed different parameters of publication bias (Begg and Mazumdar rank order correlation, Classic and the Orwin fail-safe N and Duval and Tweedie's Trim and Fill test) in order to avoid any sort of bias in our study and analysis.

·         The report and display structure were adopted PRISMA checklist in order to minimise any possibility of author opinions influencing the results or its subsequent interpretations.

·         This should allow for ease of dissemination and use as a repository of collated information for future studies as well.

14.  The discussion is equally impenetrable, and many statements are simply not correlated with the data (for instance, why the authors wrote a paragraph on “In a study on lung cancer cells, miR-17-5p downregulation was associated with an increased 282 expression of beclin 1 gene which is an autophagy modulator in the survival pathway.”? The fact that “this miRNA belongs to the miR-17-92 cluster and is up-regulated in PC where it is linked 284 to pancreatic carcinogenesis.” Does not related to its potential role in chemoresistance)

·         We would like to highlight the molecular behaviour of that particular miRNA in different cancers, thus have compared our findings with lung cancer where miR-17-5p is downregulated, but in pancreatic cancer it is upregulated, and associated with the PTEN pathway and therefore, PI3K-Akt signalling.

·         We tried to say that the same miRNA when differentially regulated (up or down) switches its pathway in different cancers and can exhibit a discordant expression effects on carcinogenesis and theragnosis.

15.  The conclusions (i.e., “This knowledge could drive the choice of chemotherapeutic regimens used in patients”) should be rewritten.

We have modified the sentence.

Reviewer 3:

Our sincere thanks for your valuable and thoughtful review. Our clinical oncology research team is pleased that your suggestions would improve the quality of our manuscript. It is greatly appreciated.

Specific comments:

16.  In Tables 1 and 2 the acronym “NM” should be defined and/or spelled out.

We have mentioned in the foot notes. (Line no: 283)

17.  There is no “call-out” to Figure 3 in the text; an improved figure legend for Figure 3 would help clarify the information presented in this figure. It is not clear what is meant by “hallmarks of pancreatic cancer.” Green and red sections denote miRNAs that are up-regulated and down-regulated, respectively; is this up- or down-regulation the basal expression of these miRNAs in pancreatic cancers compared to healthy tissue or is this in response to chemotherapy? Finally, do the gene names on the outside edge of the circle reference targets of the miRNAs or the pathways/mechanisms by which their expression is regulated in pancreatic cancer? A clearer figure legend should address and provide information to answer these questions.

We have modified the figure legends. The outside edge of the circle refers to the drug target by the miRNAs. (Line no: 310-312; 342)

18. A “call-out” to Table 2 in the text would be useful (likely inserted in paragraphs on page 15). A re-labelling of Table 2 that describes “chemosensitivity” to Table 3 would be useful, as the “chemoresistance” and “chemosensitivity” tables are indeed two separate tables (as of now both are labelled Table 2). The new Table 3 should be referenced in the text (likely in the paragraph on page 16).

We have renamed the table as 2 and 3, respectively, and listed out in the text section. (Line no: 344)

19.  An expanded comment in the discussion section on the meta-analysis of miR-21 involvement in pancreatic cancer would be useful; the authors’ analysis suggests that mIR-21 expression does not necessarily predict chemoresistance or patient survival (data in Figure 5): this is an important finding and should be discussed.

We would like to refrain from commenting heavily on miR-21 as the power of the analysis is too low to comment on the results obtained from the miR-21 subgroup. Although our results do show that miR-21 may not have any significant effect on directing patient survival or chemotherapeutic response, the fact that this conclusion was draw from a total of 2 studies precludes us from forming any concrete responses. Therefore, we have remained silent of the matter, until further assessment is conducted, and more data is available.

Many Thanks for your valuable time.

Thanking you,

Dr Jayaraj

Round  2

Reviewer 2 Report

the authors improved the quality of the figures and text of their manuscript